# Bacterial Populations in International Artisanal Kefirs

**DOI:** 10.3390/microorganisms8091318

**Published:** 2020-08-29

**Authors:** Abrar Sindi, Md. Bahadur Badsha, Gülhan Ünlü

**Affiliations:** 1Department of Animal, Veterinary, and Food Sciences, University of Idaho, Moscow, ID 83844-2330, USA; sind3904@vandals.uidaho.edu; 2Institute for Modeling Collaboration and Innovation (IMCI), University of Idaho, Moscow, ID 83844-1122, USA; mdbadsha@uidaho.edu; 3Sera Prognostics, Inc., 2749 East Parleys Way, Suite 200, Salt Lake City, UT 84109, USA; 4School of Food Science, Washington State University, P.O. Box 646376, Pullman, WA 99164-6376, USA; 5Department of Biological Engineering, University of Idaho, Moscow, ID 83844-0904, USA

**Keywords:** fermented food, fermented beverage, dairy fermentation, artisanal kefir, artisanal kefir grain, lactic acid bacteria (LAB), acetic acid bacteria, 16S rRNA gene sequencing, microbial population, microbiome, probiotic, health benefits, bacteriocin, safe food

## Abstract

Artisanal kefir is a traditional fermented dairy product made using kefir grains. Kefir has documented natural antimicrobial activity and health benefits. A typical kefir microbial community includes lactic acid bacteria (LAB), acetic acid bacteria, and yeast among other species in a symbiotic matrix. In the presented work, the 16S rRNA gene sequencing was used to reveal bacterial populations and elucidate the diversity and abundance of LAB species in international artisanal kefirs from Fusion Tea, Britain, the Caucuses region, Ireland, Lithuania, and South Korea. Bacterial species found in high abundance in most artisanal kefirs included *Lactobacillus kefiranofaciens, Lentilactobacillus kefiri,*
*Lactobacillus ultunensis*, *Lactobacillus apis, Lactobacillus gigeriorum, Gluconobacter morbifer, Acetobacter orleanensis, Acetobacter pasteurianus, Acidocella aluminiidurans,* and *Lactobacillus helveticus*. Some of these bacterial species are LAB that have been reported for their bacteriocin production capabilities and/or health promoting properties.

## 1. Introduction

Artisanal kefir is an ancient fermented beverage obtained via fermentation of milk by kefir grains [1]. Kefir grains are a combination of yeast, bacteria, and bacterial polysaccharides [2]. Up to 50 different bacterial and yeast species have been identified in artisanal kefirs [1,3,4]. Numerous combinations of these microorganisms at the species level lead to artisanal kefirs with unique characteristics. It is important to determine the specific microbial compositions of artisanal kefirs and their grains from different origins [5] to obtain a better understanding of kefir as a functional dairy product. Early attempts to isolate kefir-associated microorganisms were obstructed by these microorganisms’ fastidious nature. The lactic acid bacteria (LAB) found in kefir require specific nutrients and conditions for growth, so select LAB in kefir may have been undetected via culture-dependent methods [6]. Culture-independent methods are commonly used to identify microbial diversity in fermented foods and beverages. Among these methods, the 16S rRNA gene sequencing has been suggested as a suitable method for identification of bacteria at the species level [7].

As previously mentioned, a typical kefir microbial community includes LAB, acetic acid bacteria, and yeast among other species in a symbiotic matrix [8]. The 16S rRNA gene sequencing has been successfully used to identify bacterial species in kefir [9,10]. For example, Gulitz et al. identified *Lactobacillus nagelii* (*Liquorilactobacillus nagelii*), *Lactobacillus hordei* (*Liquorilactobacillus hordei*), *Bifidobacterium psychraerophilum, Lactobacillus hilgardii* (*Lentilactobacillus hilgardii*), *Lactobacillus satsumensis* (*Liquorilactobacillus satsumensis*), *Acetobacter orientalis, Clostridium tyrobutyricum,* and *Leuconostoc citreum* from kefir originated from Germany using the 16S rRNA gene sequencing of the V1 to V4 hypervariable regions [10]. Another study identified *Lactobacillus kefiranofaciens, Lactobacillus acidophilus,* and *Lactobacillus sunkii* (*Lentilactobacillus sunkii*) with abundance of 77–78%, 10–11%, and 2–4%, respectively, in two Turkish kefir grains using whole genome and 16S rRNA shotgun sequencing [2]. 

LAB are known to produce one or more of the following products with antimicrobial properties: organic acids, free fatty acids, diacetyl, hydrogen peroxide, and bacteriocins. Bacteriocins are ribosomally synthesized proteins or peptides that are secreted by bacteria and inhibit closely related Gram-positive and some Gram-negative bacteria using various mechanisms of action [11]. Kefir contains LAB which are known to produce bacteriocins. For example, Lacticin 3147, a bacteriocin produced by *Lactococcus lactis* DPC3147 isolated from Brazilian kefir grains, inhibited the growth of *Escherichia coli*, *Listeria monocytogenes, Salmonella typhimurium,* and *Salmonella enteritidis* [12]. 

FAO/WHO (Food and Agriculture Organization and the World Health Organization) define probiotics as “live microorganisms which when administered in adequate amounts confer a health benefit to the host” [13]. Recently, The International Scientific Association for Probiotics and Prebiotics recommended that “the term probiotic be used only on products that deliver live microorganisms with a suitable viable count of well-defined strains with a reasonable expectation of delivering benefits for the wellbeing of the host” [14]. Kefir, reported as a natural probiotic beverage by some researchers [1], contains LAB species with documented beneficial properties. For example, *L. kefiranofaciens* XL10, which is a homofermentative LAB that produces lactic acid as the main product, was reported to have probiotic properties in both in vitro and in vivo studies [15,16]. *Lactobacillus kefiri* (*Lentilactobacillus kefiri*) is a heterofermentative LAB that produces lactic acid, acetic acid, ethanol, and carbon dioxide, and was reported, in both in vivo and in vitro studies, to have probiotic properties such as adherence to mucus extracted from the small intestine and colon, strong cholesterol assimilation abilities, and lowering the secretion of IL-8 caused by *Salmonella enterica* infection [17,18].

In our recently published work [19], the antimicrobial activities of artisanal kefir products from Fusion Tea, Britain, the Caucuses region, Ireland, Lithuania, and South Korea were investigated against select foodborne pathogens. It was confirmed that bacteriocin production is the main reason for these kefirs’ antimicrobial activity [19]. In the presented work, the 16S rRNA gene sequencing was used to reveal bacterial populations and elucidate the diversity and abundance of LAB species in these international artisanal kefirs. Several LAB species identified in the presented work have been reported by other researchers to have bacteriocin production capabilities and/or health promoting properties. Based on our findings, bacteriocin-producing and/or potentially beneficial LAB are being isolated from select artisanal kefirs and being characterized in our laboratory.

## 2. Materials and Methods 

### 2.1. Genetic Approaches for Identification of Bacteria in Kefir

#### 2.1.1. Kefir Preparation

Six artisanal kefir grains originating from South Korea (K; [20]), Ireland (I; Etsy Inc.), Lithuania (L; Etsy Inc.), Britain (B; Etsy Inc.), the Caucuses (C; Etsy Inc.), and a compilation of world-sourced grains blended (A; Fusion Tea, Amazon) were used in this study. Kefir grains were inoculated (10% (w/v)) into whole pasteurized milk and incubated at 22–24 °C for 24 h. At the end of the fermentation process (pH 3.9–4.1), clean plastic strainers with 1-mm pore size was used to separate kefir grains from kefir products. Designated plastic strainers were used for each kefir product/grains to avoid cross contamination. Kefir products were centrifuged (Eppendorf Microcentrifuge 5415D, Hauppauge, NY, USA) at 16,000 *g* for 10 min at room temperature to remove the lipid layer. Kefir grains separated were washed three times with sterilized DI water. Kefir grains were mixed with sterile DI water (3 mL), homogenized using Stomacher 400 (Seward Limited, Worthing, West Sussex, UK) for 60–120 s at high speed, and then centrifuged at 16,000 *g* for 10 min at room temperature to remove the lipid layer. The resulting pellets were used for DNA extractions.

#### 2.1.2. DNA Extraction

The E.Z.N.A Universal Pathogen Kit (Omega Biotech, Norcross, GA, USA) was used for DNA extraction from all international artisanal kefirs and their grains. The kit’s user guide was followed with two additional washing steps with the DNA washing buffers included in the kit. The concentration of DNA preparations was determined by ultraviolet spectrophotometry (Spectra MAX 190). Absorbance 260/280 values between 1.8 and 2.0 were considered acceptable. The quality of DNA preparations was determined by running DNA samples on 1.2% (w/v) agarose gels with TAE (1X) buffer and using PowerPac 200 submerged horizontal gel electrophoresis systems from Bio-Rad (Hercules, CA, USA). Two DNA samples, one representing a given kefir product and another representing the kefir grains in that product, were sent out to Omega Bioservices (Norcross, GA, USA) for the 16S rRNA gene sequencing. With two DNA samples representing a given artisanal kefir, a total of 12 DNA samples representing six international artisanal kefirs were sequenced by Omega Bioservices.

Commercial kefirs, made with known cultures indicated on their labels, were purchased from local (Moscow, ID, USA) grocery stores and used as controls: Lifeway (plain), The Greek Gods (plain), Wallaby (organic plain), and Maple Hill (organic plain). DNA from the commercial kefir samples was extracted by Omega Bioservices using their E.Z.N.A Universal Pathogen Kit.

#### 2.1.3. The16S rRNA Gene Sequencing

Illumina MiSeq Sequencing was used for the 16S rRNA gene sequencing (≈200K reads per sample) by Omega Bioservices. Both the V1–V3 and the V3–V4 primer sets were used for PCR amplification by Omega Bioservices. The V1 and V2 regions have been historically used for identification of LAB [21]. All variable regions in the 16S rRNA gene have been reported to be effective in identifying bacterial communities in kefir through the 16S rRNA gene sequencing [22]. Library preparation type was KAPA HiFi PCR as per Omega Bioservices. Data was delivered via Illumina BaseSpace web site.

### 2.2. The 16S rRNA Gene Sequencing Analyses

The 16S rRNA gene sequencing reads were obtained from Omega Bioservices for the V1–V3 and the V3–V4 regions. The total number of reads (often referred to as read depth) were converted to relative abundance and rounded to 0.1%. The percent relative abundance was calculated by dividing the number of reads for each phylum, genera, or species with the total number of reads for each kefir and multiplying the outcome by 100. Pie charts were created for the relative abundances of bacterial phyla in kefir products and their grains based on the V1–V3 and V3–V4 regions of the 16S rRNA genes. Stacked bar charts were created for the average abundance of the 21 most abundant bacterial genera, based on the V1–V3 and V3–V4 regions of the 16S rRNA genes, in kefir products and their grains. Additional stacked bar charts were created for the average relative abundance of the 10 most abundant bacterial species, based on the maximum percentage between the V1–V3 and the V3–V4 regions of the 16S rRNA genes, and scaling (0–100) species contribution. Stacked bar charts were also created for the percent species contribution of LAB found in kefir samples and known to produce bacteriocins, based on the maximum percentage between the V1–V3 and the V3–V4 regions of the 16S rRNA genes, and scaling (0–40) species contribution. Canonical correlations were performed using the aggregated lists of taxa to examine relationships among kefir products and their grains. Correlation plots were used to understand whether and how strongly kefir products and their grains are related. The Pearson’s correlation coefficient (*r*) provides information about the strength and direction of a relationship between a given kefir product and its grains. The regression line describes how kefir products change as kefir grains change. Heatmaps were created using Pearson correlations among the kefir products and their grains. The results of hierarchical clustering (HC) using complete linkage distance method represented by a dendrogram was included to show taxonomic relationships among international artisanal kefirs. The Pearson’s correlation coefficients, regression lines, and dendrogram were generated by R-3.5.1 programming (R Studio Inc., Boston, MA, USA). 

## 3. Results

### 3.1. Genetic Approaches for Identification of Bacteria in Kefir Samples

#### 3.1.1. Bacterial Phyla and Genera Present in Artisanal Kefirs and Their Grains

Based on the 16S rRNA sequencing data generated in this work, a diverse group of bacteria were determined to be present in artisanal kefir products and their grains from different regions of the world. Specifically, relative abundances of bacterial phyla and bacterial genera based on the V1–V3 and the V3–V4 regions of the 16S rRNA genes were determined for the kefir product from Lithuania (LP), kefir grains from Lithuania (LG), kefir product from South Korea (KP), kefir grains from South Korea (KG), kefir product from Ireland (IP), kefir grains from Ireland (IG), kefir product from the Caucuses region (CP), kefir grains from the Caucuses region (CG), kefir product from Britain (BP), kefir grains from Britain (BG), kefir product from Fusion Tea (AP, Amazon), and kefir grains from Fusion Tea (AG, Amazon) (Figure 1). The *Firmicutes* phylum was the most abundant in all artisanal kefir products, followed by the phyla *Proteobacteria, Actinobacteria, Verrucomicrobia, Planctomycetes,* and *Nitrospirae* (Figure 1). The phylum *Firmicutes* was the most abundant phylum in kefir grains, especially for CG, with 97.9% (based on V1–V3) and 97.7% (based on V3–V4) (Figure 1), followed by BG, LG, AG, KG, and IG for the V1–V3 regions (Figure 1). Based on the V3–V4 regions, *Firmicutes* was the most abundant phylum in CG followed by LG, BG, AG, KG, and IG (Figure 1). The following genera were identified as the major genera in artisanal kefirs: *Lactobacillus*, *Lentilactobacillus, Lacticaseibacillus, Acetobacter, Swaminathania*, *Gluconobacter*, *Streptococcus*, *Pediococcus, Pseudomonas, Acidocella, Cohnella, Peptoniphilus, Saccharopolyspora, Thermodesulfovibrio, Singulisphaera, Chthoniobacter, Paenibacillus, Knoellia, Leuconostoc, Bifidobacterium,* and *Lactococcus* (Figure 1).

The 16S rRNA gene sequencing targeting the V1–V3 and the V3–V4 regions resulted in various relative abundance percentages for the phylum and genus level designations in all artisanal kefirs (Figure 1). For example, LP has 67% *Firmicutes* based on the V1–V3 region, while it has 64.6% *Firmicutes* based on the V3–V4 region. The IP has 65.6% and 69.7% *Firmicutes* based on the V1–V3 and the V3–V4 regions, respectively. The AP has the highest relative abundance for *Lactobacillus* species, from 75% (based on V1–V3) to 75.5 % (based on V3–V4) while LP has the lowest abundance for *Lactobacillus* species from 61.6% (based on V3–V4) to 64.2% (based on V1–V3). The genus *Lentilactobacillus* was found to be in the highest abundance in IG, from 7.7% (based on V1–V3) to 7.4% (based on V3–V3), when compared to all other kefir grains and products. It appears that the combination of both the V1–V3 and V3–V4 regions to identify bacteria at the phylum and genus levels in kefir products and their grains worked well. As an exception, the V3–V4 region identified the genus *Swaminathania* in all kefir products and in all kefir grains (Figure 1B), while it was not identified by the V1–V3 region (Figure 1A). 

Commercial kefirs are defined, prepared using a starter culture of LAB and yeast species as indicated on the packaging. Therefore, the 16S rRNA gene sequencing was performed on commercial kefirs as controls to test the accuracy of the identification. Commercial kefirs used in the study were determined to have the exact probiotic bacteria, both at the genus and species level as listed on their labels. (Appendix A).

#### 3.1.2. Bacterial Species Present in Artisanal Kefirs and Their Grains 

The 10 most abundant bacterial species found in artisanal kefirs included L. kefiranofaciens, Lent. kefiri, Lactobacillus ultunensis, Lactobacillus apis, Lactobacillus gigeriorum, Gluconobacter morbifer, Acetobacter orleanensis, Acetobacter pasteurianus, Acidocella aluminiidurans, and Lactobacillus helveticus (Figure 2). L. kefiranofaciens was determined to be the most abundant bacterium in all artisanal kefirs with relative abundance between 48.22% and 93.76% (Figure 2). The relative abundance for Lent. kefiri varied between 2.94% (KP) and 7.3% (AP) among kefir products and between 3.7% (AG) and 7.86% (IG) among kefir grains (Figure 2). L. ultunensis was found to be more abundant in KP and KG when compared to all other artisanal kefirs and their grains. A. pasteurianus was more abundant (17.1%) in KP while G. morbifer was more abundant (5.8%) in LP when compared to their relative abundance in all other artisanal kefirs.

Differences exist in artisanal kefirs regarding microorganisms present in kefir products versus their grains. For example, *L. kefiranofaciens* exhibits a higher relative abundance in all kefir grains when compared to their corresponding products (Figure 2). Similarly, *Lent. kefiri* exhibits a higher relative abundance in kefir grains B, I, K, and L when compared to their corresponding products (Figure 2). On the other hand, *Lent. kefiri* exhibits a lower relative abundance (3%) in AG when compared to that (6%) of in AP (Figure 2). 

#### 3.1.3. Bacteriocinogenic and Beneficial Bacteria in Artisanal Kefirs

Several LAB species recognized as bacteriocin producers in the current literature were identified in all artisanal kefirs subjected to this work, albeit in various relative abundance (Figure 3). The following bacteriocinogenic species were found in artisanal kefirs: *Lent. kefiri, L. helveticus, Lactobacillus delbrueckii, Lacticaseibacillus paracasei (Lactobacillus paracasei), Lacticaseibacillus casei (Lactobacillus casei), Lacticaseibacillus rhamnosus (Lactobacillus rhamnosus), L. apis, Lactobacillus crispatus, Lactobacillus acidophilus*, *Streptococcus thermophilus, Leuconostoc mesenteroides,* and *Lactococcus lactis* (Figure 3). All artisanal kefirs, except kefir K, were determined to contain *Lent. kefiri* as the most abundant LAB with known bacteriocin production capability. *Lent. kefiri* was the third most abundant LAB with known bacteriocin production in Kefir K (Figure 3).

Kefir K was a stand-alone kefir product with respect to the ranking of species with known bacteriocin production: *L. helveticus, L. crispatus, Lent. kefiri, L. apis,* and *L. acidophilus*. The most abundant LAB with known bacteriocin production capability was identified to be *L. helveticus* in KP with relative abundance of 21.8% and in KG with relative abundance of 17.8% (Figure 3). *L. helveticus* was found in other artisanal kefir products with relative abundance between 0.26% (BP) and 4.11% (CP) and in artisanal kefir grains with relative abundance between 0.13% (LG) and 3.34% (IG).

*Lactobacillus crispatus*, another *Lactobacillus* species with known bacteriocin production, followed *L. helveticus* in KP and KG with relative abundance of 3.44% and 2.93%, respectively (Figure 3). The relative abundance for *L. crispatus* in other artisanal kefir products and kefir grains varied between 0.07% (BP) and 0.6% (CP) and between 0.045% (BG) and 0.51% (IG), respectively. 

*Lactobacillus apis*, a lesser known *Lactobacillus* species with reported bacteriocin production capability, was determined to be present in all artisanal kefir products and their grains. The relative abundance of *L. apis* was in the range of 1.79% (LP) to 2.17% (IP) in kefir products and 2.21% (CG) to 2.54% (AG) in kefir grains (Figure 3). *L. apis* was determined to have the second highest and the third highest relative abundance among other bacteriocin producing LAB in kefir products A, B, I, and L and kefir product C, respectively (Figure 3). The organism had the second highest abundance ranking among other bacteriocin producing LAB in kefir grains A, B, C, L, while the ranking was the third highest abundance in kefir grain I and the fourth highest abundance in kefir K (Figure 3).

*Lactobacillus acidophilus*, an organism with reported bacteriocin production capability, ranked fifth most abundant bacteriocinogenic LAB species in all artisanal kefir products. *L. acidophilus* was present in KP with relative abundance of 0.7% and in KG with relative abundance of 0.5% (Figure 3). All other artisanal kefirs (A, B, C, I, and L), both products and their grains, were determined to contain much less *L. acidophilus* with relative abundance of 0.01%–0.1% (Figure 3). 

Some LAB species with known bacteriocin production capability were absent in select kefir products plus their grains (Figure 3): *S. thermophilus* in kefir A; *L. delbrueckii* in kefir A; *Lacti. paracasei* in kefirs A and B; *Lacti. casei* in kefirs A, B, C, I, and K; *Lacti. rhamnosus* in kefir K; *Lac. lactis* in kefirs B, C, and K; and *Leu. mesenteroides* in kefirs A, B, I, and L. *Lacti. casei* was determined to be present, albeit at a low relative abundance, in kefir L grains only (Figure 3).

#### 3.1.4. Correlation of Artisanal Kefir Products to Their Grains and Taxonomic Relationships Among International Artisanal Kefirs

Kefir grains showed a high correlation to their products in their species content (Figure 4A–C). Kefir A presented the highest correlation between its grains and the product (*r* = 0.994), followed by kefirs I, C, B, and L with the following *r* values, respectively: *r* = 0.986, 0.986, 0.981, and 0.976. The lowest correlation was observed in kefir K (*r* = 0.948), (Figure 4B). The heatmap (Figure 4C) created confirmed that KG and KP had the lowest correlation with each other and the lowest correlation with all other kefirs (Figure 4C). The AP, on the other hand, appeared to have the highest correlation with all kefir grains (*r* = 1) except for KG. The AG was not highly correlated with other kefir products (Figure 4C). 

Hierarchical Clustering (HC) dendrogram (Figure 5) shows taxonomic relationships among international artisanal kefirs. KP appears to be a stand-alone kefir product with respect to species- level taxonomy. KP has its own highest cluster, while AP is the most related kefir to KP when compared to all other kefir products. In addition, IP and CP are related with respect to species-level taxonomy similar to LP and BP which are also related to each other (Figure 5).

## 4. Discussion

In the presented work, the 16S rRNA gene sequencing was successfully applied to six international artisanal kefirs and revealed microbial populations at the phyla, genera, and species levels in international artisanal kefirs. A total of six phyla were identified, *Firmicutes Proteobacteria, Actinobacteria, Verrucomicrobia, Planctomycetes,* and *Nitrospirae*, in artisanal kefirs studied. To our knowledge, the presence of a combination of six phyla has not been reported for any artisanal kefir. This could be due to the complexity of the artisanal kefirs examined in our work and/or the effectiveness of the 16S rRNA gene sequencing and the sequence analyses. The phylum *Firmicutes* was the most abundant phylum among all other phyla in kefir grains and their products, which is a commonly found phylum in artisanal kefir. *Lactobacillus* was the most abundant genus with the highest relative abundance in AG and the lowest relative abundance in IP. This aligns well with relative abundance of lactobacilli in artisanal kefirs described in the literature. The genus *Lactobacillus* has been recently reclassified into 25 new genera including *Lactobacillus, Lentilactobacillus,* and *Lacticaseibacillus* [23]. This reclassification was fully considered in our work. 

Our goal was to utilize multiple variable regions of the 16S rRNA gene to successfully identify all genera and species present in international artisanal kefirs. One interesting outcome of our study was the identification of the genus *Swaminathania* through the use of the V3–V4 region of the 16S rRNA gene but not through the use of the V1–V3 region. Since the 16S rRNA gene sequencing of four commercial kefirs, used as controls to test the accuracy of the identification, determined the exact probiotic bacteria listed on their labels at the species level, we are confident with the results of the 16S rRNA gene sequencing. The interesting outcome can be attributed to the fact that different variable regions in the 16S rRNA gene can be less or more suitable to identify bacteria at the genus or species level. Perhaps the V1–V3 region is less suitable for identification of the genus *Swaminathania* than the V3–V4 region. 

A study by Marsh et al. for kefirs sourced from the United Kingdom, Canada, and the United States reported that the 16S rRNA reads for the V4–V5 region identified three bacterial phyla: *Actinobacteria, Firmicutes,* and *Proteobacteria* [24]. Both *Proteobacteria* and *Firmicutes* were verified to be the most abundant phyla. The *Proteobacteria* phylum was greater in abundance, in general, in the grains than the products for all kefirs studied. The phylum *Firmicutes* was in higher abundance in the product than in the grains. The abundance of *Actinobacteria* was low, in general, in both grains and products. Marsh et al. found that the most dominant genus was *Zymomonas* with a relative abundance of 87–49% followed by *Lactobacillus,* ranging in relative abundance from 38.8% to 12% [24]. Differences observed between our work and that of Marsh et al. can be attributed to differences among the kefirs used and the regions that were targeted in the 16S rRNA sequencing. 

*Lentilactobacillus kefiranofaciens* was the most abundant lactobacilli in all artisanal kefirs. Our findings align with that of Wang et al., who reported *L. kefiranofaciens* as the sole dominant and stable species in Tibetan kefir [25]. *L. kefiranofaciens* has been reported to have probiotic properties and health benefits [26,27]. A study with both in vivo and in vitro components suggested *L. kefiranofaciens* M1 to be applied in fermented dairy products as an alternative therapy for intestinal disorders [26]. An increase in the production of regulatory T-cell cytokines was observed when *Lb. kefiranofaciens* M1 was cocultured with spleen cells [27]. 

Other highly abundant bacterial species identified in artisanal kefirs in the presented work included the following organisms: *Lenti. kefiri, L. ultunensis*, *L. apis, L. gigeriorum, G. morbifer, A. orleanensis, A. pasteurianus, Acid. aluminiidurans,* and *L. helveticus.* Out of these, *L. ultunensis* and *L. apis* were reported to be present in kefir grains from different regions of Turkey [28]. Based on our knowledge, *L. gigeriorum* has not been reported in kefir until the presented work, however, the organism has been reported to be closely related to *L. acidophilus* [29]. In addition, our work is the first study reporting *Acid. aluminiidurans* occurrence in kefir. *Acid. aluminiidurans* is an aluminum-, acid-, and sulfate-tolerant bacterium, which was originally isolated from a waterweed in Vietnam [30].

*Lentilactobacillus kefiri* was found in higher relative abundance in kefir grains B, I, K, and L when compared to their corresponding products. Perhaps *Lenti. kefir* is imbedded in the outer layers of the kefir A grains and immigrate, during kefir fermentation, more into the kefir product than in kefirs B, I, K, and L. A study by Korsak et al. evaluated the microbiota of kefir samples from Belgium using 16S pyrosequencing revealed the presence of *L. kefiranofaciens, Lactococcus lactis* ssp*. cremoris*, *Gluconobacter frateurii*, *Lenti. kefiri*, *A. orientalis*, *Leu. mesenteroides,* and *Acetobacter lovaniensis* [31]. In that study, some samples showed *L. kefiranofaciens* to be the most abundant—similar to our work—while other samples showed *Lac. lactis* as the most abundant, constituting ≈80% of the bacterial population. Korsak et al. samples [31] did not report the occurrence of *L. ultunensis*, *L. apis, L. gigeriorum, G. morbifer, A. orleanensis, A. pasteurianus, Acid. aluminiidurans,* and *L. helveticus*, which are being reported in the presented study. As expected, similarities and differences exist among international artisanal kefirs regarding their bacterial populations at the species level. 

Acetic acid bacteria have been reported to contribute to exopolysaccharide formation and increase in kefir grain biomass, without negatively affecting the sensory properties and other microflora of kefir [32]. *Acetobacter fabarum*, *A. lovaniensis,* and *A. orientalis* were identified in kefir originated from Italy [33]. *A. orleanensis*, *A. orientalis, Acetobacter malorum,* and *A. pasteurianus* were found in all artisanal kefirs in the presented work. Therefore, it appears that different acetic acid bacteria species occupy different artisanal kefirs. The genus *Gluconobacter* is a member of the acetic acid bacteria family. Research has indicated a symbiotic relationship between LAB isolated from kefir and *Gluconobacter* spp. [34]. *G. frateuii* was identified in kefir originated from Belgium [31]. *G. morbifer* and *Gluconobacter kondonii* were identified in all artisanal kefirs examined in the presented work. 

Kefir has been reported to be a health-promoting beverage. Numerous studies have suggested kefir’s health benefits in terms of improving lactose digestion [35], protecting against foodborne pathogens, anticancer effects [36], immunomodulatory effects [37], and probiotic activity [5]. These health benefits appear to be related to the kefir microbiota and/or their metabolites [5].

Potential probiotics need to exhibit functional properties such as viability and persistence in the GI-tract, immunomodulation, antagonistic and antimutagenic properties [38]. The antagonistic abilities of probiotics include aggregation and coaggregation, adhesion to the intestine, inhibiting pathogenic bacterial adhesion to the intestine as well as production of antimicrobial substances such as bacteriocins [39]. Occurrence of LAB with probiotic potential in kefir has been reported. *L. acidophilus* LA15, *Lactobacillus plantarum* (*Lactiplantibacillus plantarum*) and *L. kefiri* (*Lentilactobacillus kefiri*) D17 isolated from Tibetan kefir were proposed as beneficial probiotics [17]. *L. rhamnosus* (*Lacticaseibacillus rhamnosus*) is reported to be a probiotic organism found in kefir [40]. *L. paracasei* (*Lacticaseibacillus paracasei*) MRS59 displayed significant antioxidant activity and adhesion to Caco-2 cells, which indicated its probiotic potential [41]. *L. kefiranofaciens* 8U, *Lactobacillus diolivorans* (*Lentilactobacillus diolivorans*) 1Z, and *L. casei* (*Lacticaseibacillus casei*) 17U isolated from Brazilian kefir were reported as potential probiotics [5]. Moreover, 11 *Lac. lactis* strains isolated from Brazilian kefir were reported to show probiotic properties such as antagonistic activity and antioxidative activity [41]. *L. acidophilus* Z1L, *L. helveticus* Z5L, and *L. casei* (*Lacticaseibacillus*
*casei*) Z7L, isolated from Turkish homemade kefirs, were reported to have probiotic activities [42]. *L. helveticus* was identified in all artisanal kefirs in the presented work. Our results indicated that *L. helveticus* in kefir K product and kefir K grains was 5–84 times and 5–137 times more abundant than other artisanal kefirs and artisanal kefir grains, respectively. In this study, all bacteria found in kefir with known bacteriocin producing capabilities were reported to have beneficial properties, except *L. apis*. To our knowledge, *L. apis* has not been reported as a beneficial bacterium. *L. plantarum* Lp27, isolated from Tibetan kefir, exhibited efficient cholesterol-reducing ability [43]. *L. plantarum* was not found in any kefir grains or products in the current study.

The 16S rRNA gene sequencing described in the presented work has allowed us to determine LAB that might be responsible for the production of bacteriocins which were linked to the inhibition of foodborne pathogenic bacteria in our previously published study [19]. *Lenti. kefiri* is a bacterium that has been isolated from kefir and shown to inhibit both Gram (+) and Gram (-) pathogens [44,45]. *Lenti. kefiri* was the second most abundant lactobacilli in all artisanal kefirs except kefir K. In Taiwanese kefir grains, *Lenti. kefiri* was determined to be the most abundant *Lactobacillus* species [46,47]. *L. helveticus,* a species known to produce bacteriocins, was found to be the most abundant lactobacilli in kefir K. This bacterium was reported to produce heat-labile, large molecular mass (>30 kDa) peptides lysostaphin, enterolysin A, and helveticin J with antimicrobial activities [48,49]. *L. acidophilus* was reported to affect the membrane permeability and cell wall formation of its target organisms by producing acidocin B, entereocin P, and reuterin 6 peptides [50,51]. *L. crispatus* was reported in both in vivo and clinical studies to have antimicrobial activity against bacterial vaginosis and uropathogenic *Escherichia coli* [52,53]. In the presented work, *L. crispatus* was found to be 5.7–49 times and 5.7–65 times more abundant in KP and KG, respectively, when compared to other artisanal kefir products and their grains. *Leu. mesenteroides* was found in kefirs K and C in our work. The organism was reported to produce Leucocyclicin Q, a novel cyclic bacteriocin which shows antimicrobial activity against Gram-positive bacteria such as *Bacillus coagulans* [54]. *Lacti. casei,* which was found only in kefir L in the presented work, was reported to be effective against *Lis. monocytogenes, Listeria innocua, Corynebacterium difterium,* and *Bacillus cereus*. *Lacti. rhamnosus* was found in small abundance in kefir C, I, and L in our work. This organism was reported to inhibit *Staphylococcus aureus, Lis. monocytogenes, Lis. innocua, C. diphtheriae,* and *B. cereus* [55]. 

A research project that focused on the microbial diversity of Tibetan kefir grains from different origins did not detect *Lac. lactis* in kefir grains examined but in kefir products [56,57]. Another published study found that *Lac. lactis* and *S. thermophilus* were dominant microorganisms accounting for 53–65% of the total microflora of Tibetan kefir grains and accounting for 74–86% of the total microflora of kefir products [58]. The artisanal kefirs tested in our work exhibit very low relative abundance for these organisms and thus they do not resemble the Tibetan kefir with dominant *Lac. lactis* and *S. thermophilus*. The variation in bacterial distribution in kefir products versus their grains can be attributed to temperature increase created by active fermentation or where these bacteria exist in the kefir grain [4] among other factors.

Owing to the fact that kefir A is a mixture of grains from various geographical regions, it is not surprising that it is related to kefir K and to all other kefirs in species level taxonomy. An interesting result for the species level taxonomy is that the Irish kefir is not closely related to the British kefir—even though Ireland is geographically close to Britain—but it is related to the Caucuses kefir. The British kefir is found to be closely related to the Lithuanian kefir. Kefir K, on the other hand, appears to be the most unique kefir in terms of its species-level taxonomy and its composition comprised of LAB reported to produce bacteriocins. In our former work [19], kefir K exhibited antimicrobial activity against a diverse group of foodborne pathogenic indicators. Due to kefir K’s robust antimicrobial activity and its unique species-level taxonomy, our goal is to carry out additional research on kefir K with emphasis on isolation and characterization of LAB and their bacteriocins as well as application of these bacteriocins as natural, clean-label biopreservatives for shelf life protection and assurance of microbial food safety.

## 5. Conclusions

Geographical origins of kefir grains and kefir production methods affect the microbial composition of artisanal kefirs. Types of milk, incubation temperatures, incubation times, and the ratios of kefir grains to milk play important roles on kefir’s microbial composition [1,59,60]. Kefir grains have been shown to exhibit regional differences in microbial composition due in part to local LAB finding a niche in the grains [6]. The culture-independent method employed in our work, the 16S rRNA gene sequencing, successfully revealed the microbial populations in six international artisanal kefirs and demonstrated the diversity and abundance of LAB found in each kefir tested, many with reported capability of producing bacteriocins and potential health benefits. Species found in high relative abundance in most artisanal kefirs included *L. kefiranofaciens, Lenti. kefiri, L. ultunensis*, *L. apis, L. gigeriorum, G. morbifer, A. orleanensis, A. pasteurianus, Acid. aluminiidurans,* and *L. helveticus*. LAB with documented bacteriocin production capabilities, *Strep. thermophilus, Lenti. kefiri, L. helveticus, L. delbrueckii, Lacti. paracasei, Lacti. casei, Lacti. rhamnosus, L. crispatus, Leu. mesenteroides, L. acidophilus,* and *Lac. lactis*, were found in diverse relative abundances in the artisanal kefirs examined in this study. LAB species with documented health benefits in the literature and identified in the artisanal kefirs tested in this work were *Lent. kefiri, L. helveticus, L. delbrueckii, Lacti. paracasei, Lacti. casei, Lacti. rhamnosus, L. crispatus, L. acidophilus*, *S. thermophilus, Leu. mesenteroides,* and *Lac. lactis.*


## Figures and Tables

**Figure 1 microorganisms-08-01318-f001:**
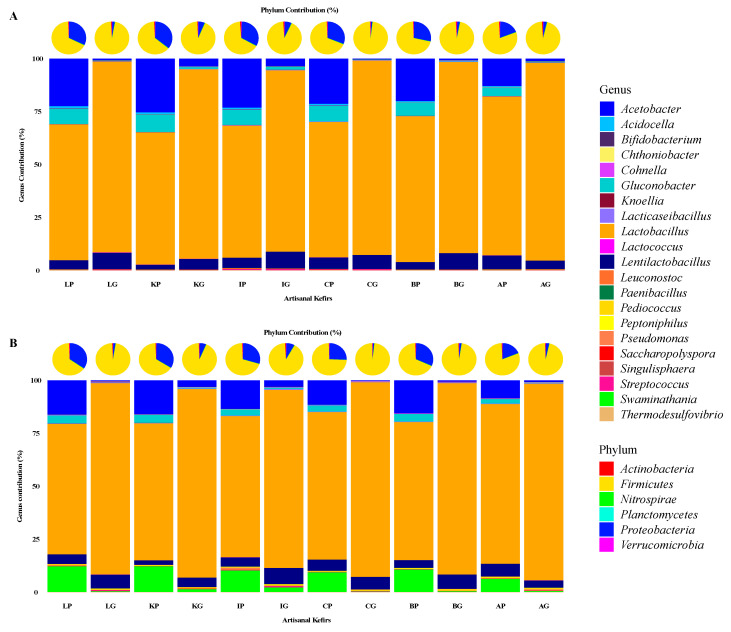
Relative abundances of bacterial phyla (pie charts) and an aggregation of the 21 most abundant bacterial genera (column charts) for the V1–V3 (**A**) and V3–V4 (**B**) regions of the 16S rRNA genes. Kefir product from Lithuania (LP), kefir grains from Lithuania (LG), kefir product from South Korea (KP), kefir grains from South Korea (KG), kefir product from Ireland (IP), kefir grains from Ireland (IG), kefir product from the Caucuses region (CP), kefir grains from the Caucuses region (CG), kefir product from Britain (BP), kefir grains from Britain (BG), kefir product from Fusion Tea (AP, Amazon), kefir grains from Fusion Tea (AG, Amazon).

**Figure 2 microorganisms-08-01318-f002:**
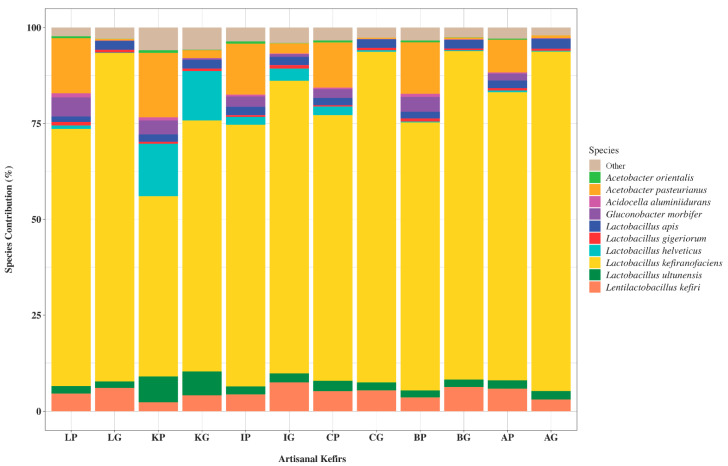
An aggregation of the 10 most abundant bacterial species, using the calculated relative abundance and scaling (0–100) species contribution. Kefir product from Lithuania (LP), kefir grains from Lithuania (LG), kefir product from South Korea (KP), kefir grains from South Korea (KG), kefir product from Ireland (IP), kefir grains from Ireland (IG), kefir product from the Caucuses region (CP), kefir grains from the Caucuses region (CG), kefir product from Britain (BP), kefir grains from Britain (BG), kefir product from Fusion Tea (AP, Amazon), kefir grains from Fusion Tea (AG, Amazon).

**Figure 3 microorganisms-08-01318-f003:**
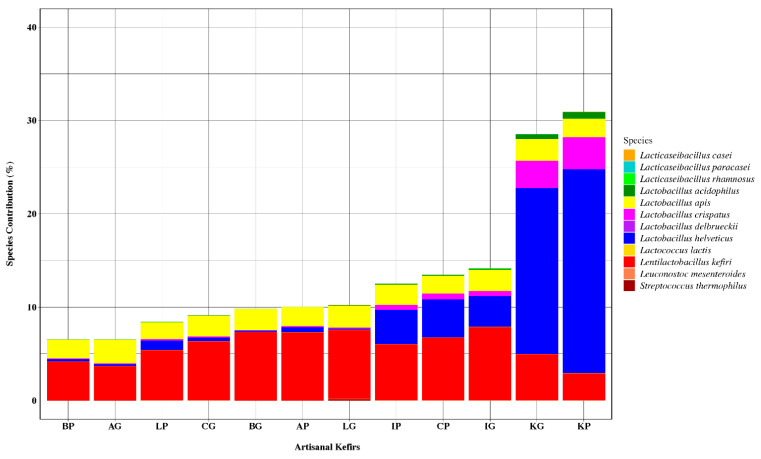
The 16S rRNA gene sequencing results showing percent species contribution of lactic acid bacteria (LAB). The IP, and IG share the same ranking with CP: *L. apis*, *L. helveticus, L. acidophilus,* and *L. crispatus.* The CG shares with IP, IG, and CP the same ranking for *L. crispatus* and *L. acidophilus* only. The LG has the highest relative abundance for *S. thermophilus* and the lowest relative abundance for *L. helveticus* when compared to LP as well as other artisanal kefirs.

**Figure 4 microorganisms-08-01318-f004:**
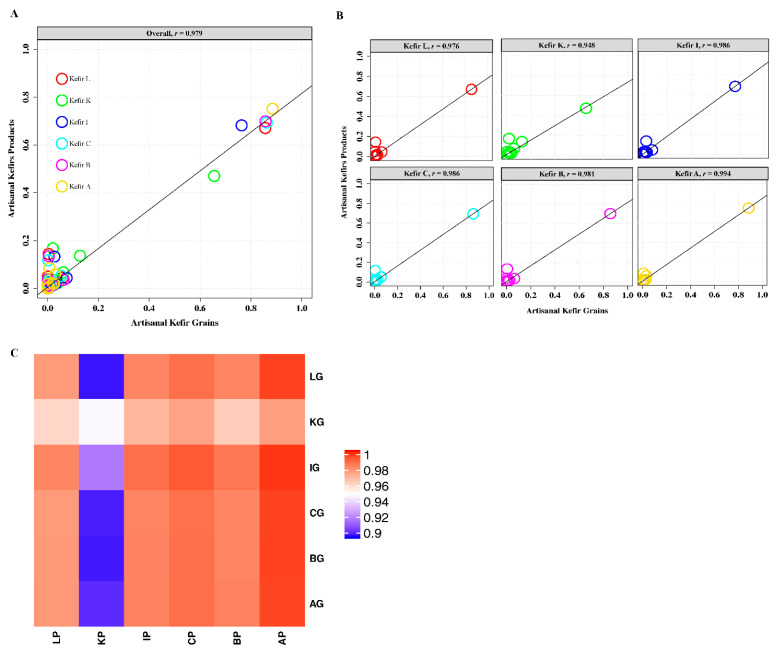
(**A**) Canonical correlation plot with a regression line between all kefir grains (x-axis for each plot) from Lithuania (LG) and South Korea (KG), Ireland (IG), the Caucuses region (CG), Britain (BG), and Fusion Tea (AG, Amazon), and all kefir products (y-axis for each plot) from Lithuania (LP) and South Korea (KP), Ireland (IP), the Caucuses region (CP), Britain (BP), and Fusion Tea (AP, Amazon) for species level. (**B**) Correlation plots with a regression line between each kefir grains and their products. The *r* indicates the Pearson’s correlation coefficient for each plot. (**C**) Heatmap showing Pearson correlation between each kefir grains and their products.

**Figure 5 microorganisms-08-01318-f005:**
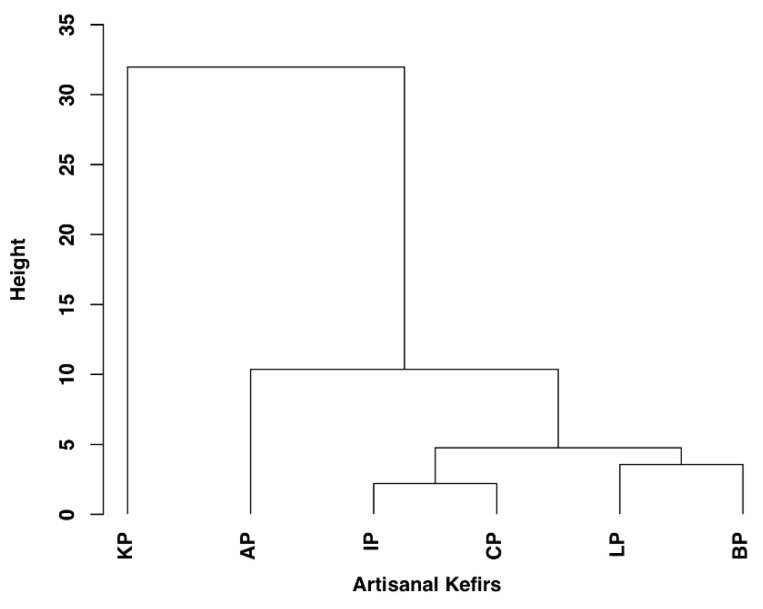
The hierarchical clustering (HC) based dendrogram showing species level taxonomy relatedness for all kefir products (X-axis) from Lithuania (LP) and South Korea (KP), Ireland (IP), the Caucuses region (CP), Britain (BP), and Fusion Tea (AP, Amazon). The complete linkage method used for computing distance between clusters to determine similar clusters.

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
