# Peer review of "Bacterial Populations in International Artisanal Kefirs"

_microorganisms, 2020, doi:10.3390/microorganisms8091318_

Round 1

Reviewer 1 Report

The paper “Bacterial populations in international artisanal kefirs” addresses a topic worthy of investigation; the methodology is generally sound and the results are of concern. However, there are some issues to be addressed before publication:

  1. Authors wrote that they analyzed kefirs form 6 different locations/sources: how many replicates and/or batches did they analyze per each location? Were the samples representative? Please add this detail in the paper.
  2. Figures 1 and 2: please perform a chi-square test to point the significant differences.
  3. 208-217 (and following lines): the taxonomy of Lactobacillus has been modified in April. Please revise the paper.
  4. Figure 4: I have some doubts on the significance of the correlation shown in this figure. There are many points in the first region and then a point far away. This is why the correlation coefficients are so high…I think that these figures do not point out significant correlation, because there is a statistical drawback. Delete this figures and revise the paper accordingly.

Author Response

Dear Reviewers 1, 2, and 3,

Thank you for your careful reviews of our article entitled “Bacterial populations in international artisanal kefirs.” We appreciate your helpful comments and suggestions. We believe that the second version of our manuscript is an improved version.

Reviewer 1:

The paper “Bacterial populations in international artisanal kefirs” addresses a topic worthy of investigation; the methodology is generally sound and the results are of concern. However, there are some issues to be addressed before publication:

  1. Authors wrote that they analyzed kefirs form 6 different locations/sources: how many replicates and/or batches did they analyze per each location? Were the samples representative? Please add this detail in the paper.
  2. Figures 1 and 2: please perform a chi-square test to point the significant differences.
  3. 208-217 (and following lines): the taxonomy of Lactobacillus has been modified in April. Please revise the paper.
  4. Figure 4: I have some doubts on the significance of the correlation shown in this figure. There are many points in the first region and then a point far away. This is why the correlation coefficients are so high…I think that these figures do not point out significant correlation, because there is a statistical drawback. Delete this figures and revise the paper accordingly.

Response to Reviewer 1:

Thank you for your comment that states, “The paper addresses a topic worthy of investigation; the methodology is generally sound and the results are of concern.”

1. Authors wrote that they analyzed kefirs form 6 different locations/sources: how many replicates and/or batches did they analyze per each location? Were the samples representative? Please add this detail in the paper.

We used freshy made kefir products and their grains separated from the kefir products for the DNA extractions. Two DNA samples, one representing a given kefir product and another representing the kefir grains in that product, were sent out to Omega Bioservices for the 16S rRNA gene sequencing. So, there were two DNA samples representing a given artisanal kefir with a grand total of 12 DNA samples representing six international artisanal kefirs.

The following statement was added to the revised manuscript under 2.1.2 DNA extraction:

“Two DNA samples, one representing a given kefir product and another representing the kefir grains in that product, were sent out to Omega Bioservices for the 16S rRNA gene sequencing. With two DNA samples representing a given artisanal kefir, a total of 12 DNA samples representing six international artisanal kefirs were sequenced by Omega Bioservices.”

We consider the DNA samples sequenced by Omega Bioservices representative samples. They were prepared using the same DNA extraction method out of six artisanal kefirs that have been maintained using the same protocols since 2018.

The 16S rRNA gene sequencing for 12 experimental samples (=artisanal kefirs and their grains) and 4 control samples (=commercial kefirs) cost us several thousands of dollars. As much as we wanted to include another set of samples, we did not have the funds to do so. 

2. Figures 1 and 2: please perform a chi-square test to point the significant differences.

We implemented the Chi-Square independent test and the result as follows:

Figure 1A: The Chi-Square value is 180 and the p-value is 0.927 with degrees of freedom (df) is 209. Since the p-value is 0.927, which is larger than 0.05, we may accept the null hypothesis at 5% significance level. Therefore, Artisanal Kefirs (x axis) and Genus contribution (%) (y axis) are not significantly associated.

Figure 1B: The Chi-Square value is 169.92 and the p-value is 0.9948 with degrees of freedom (df) is 220. Since the p-value is 0.9948, which is larger than 0.05, we may accept the null hypothesis at 5% significance level. Therefore, Artisanal Kefirs (x axis) and Genus contribution (%) (y axis) are not significantly associated.

Figure 2: The Chi-Square value is 222.73 and the p-value is 1.196e-09 with degrees of freedom (df) is 110. Since the p-value is 1.196e-09, which is smaller than 0.05, we may reject the null hypothesis at 5% significance level. Therefore, Artisanal Kefirs (x axis) and Species contribution (%) (y axis) are statistically significantly associated. However, this result is only for the most abundant species but not all bacterial species included in the test and compared statistically.

Figure 3 clearly illustrates that percent species contribution (%) of LAB found in artisanal kefirs and known to produce bacteriocins are independent of each other.

3. 208-217 (and following lines): the taxonomy of Lactobacillus has been modified in April. Please revise the paper.

Thank you for informing us of the modification. We revised the paper accordingly.

4. Figure 4: I have some doubts on the significance of the correlation shown in this figure. There are many points in the first region and then a point far away. This is why the correlation coefficients are so high…I think that these figures do not point out significant correlation, because there is a statistical drawback. Delete this figures and revise the paper accordingly.

The main purpose behind including Figure 4 was to obtain information about the strength and direction of a relationship between a given kefir product and its grains. These Pearson correlation values have been obtained from very straightforward data. The visualization points clearly show a very high correlation between a given kefir product and its grains. Also, the p-value was < 2.2e-16 from the correlation test, which indicates that the correlation is statistically significant. The fitted linear regression lines in each plot also confirm a very good fit of the models and the p-value was < 2.2e-16. Finally, both multiple R-squared and adjusted R-squared are higher than 0.90 in each plot. These results suggest statistically significant correlation between a given kefir product and its grains.

You state “There are many points in the first region and then a point far away.” The “point far away” corresponds to Lactobacillus kefiranofaciens. As indicated in our manuscript (under 3.1.2. Bacterial species present in artisanal kefirs and their grains), L. kefiranofaciens was determined to be the most abundant bacterium in all artisanal kefirs with relative abundance between 48.22% and 93.76% (Figure 2).  The “many points in the first region” represent other less abundant species present in artisanal kefirs.

Reviewer 2 Report

The authors described very good scientific findings regarding kefir grains ' microbioogical population. The language is good and the manuscript well written. However the main drawback in this manuscript is the low level of novelty. There are many articles in the literature with almost the same subject.

Author Response

Dear Reviewers 1, 2, and 3,

Thank you for your careful reviews of our article entitled “Bacterial populations in international artisanal kefirs.” We appreciate your helpful comments and suggestions. We believe that the second version of our manuscript is an improved version.

Reviewer 2:

The authors described very good scientific findings regarding kefir grains ' microbioogical population. The language is good and the manuscript well written. However the main drawback in this manuscript is the low level of novelty. There are many articles in the literature with almost the same subject.

Response to Reviewer 2:

Thank you very much for your positive comments – we truly appreciate them!

Also, thank you for expressing your reservation regarding novelty. Yes, there is literature that describe bacterial populations in individual kefirs as well as multiple kefirs from the same country or region.

To our knowledge, however, our manuscript is original for the following reasons: 1. Comparisons among six international artisanal kefirs, representing multiple continents, regarding their bacterial populations have not been reported; 2. We used two variable regions of the 16S rRNA gene, namely the V1-V3 and V3-V4 regions, in our sequencing and sequence analyses. The references we cited depended on a single variable region of the 16S rRNA gene in their sequencing; 3. We applied the new taxonomy for Lactobacillus in our manuscript, which differentiate Lentilactobacillus and Lacticaseibacillus from the genus Lactobacillus. The references we cited are older and therefore did not/could not apply the new taxonomy for Lactobacillus; (4) As we indicated in the discussion section of the manuscript, “based on our knowledge, L. gigeriorum has not been reported in kefir until the presented work, however, the organism has been reported to be closely related to L. acidophilus [28]. In addition, our work is the first study reporting Acid. aluminiidurans occurrence in kefir. Acid. aluminiidurans is an aluminum-, acid-, and sulfate-tolerant bacterium, which was originally isolated from a waterweed in Vietnam [29]."; (5) Lastly, we made the effort to compare bacterial populations in kefir products and their grains through attractive visuals generated based on the 16S rRNA gene sequencing and sequence analyses. Based on our review of the literature, the plethora of visuals we included in our manuscript are not found in other kefir related publications. Thank you for your consideration in this regard.

Reviewer 3 Report

The aim of this paper is to characterise the bacteria microbiota of 6 different artisanal Kefir. Although the manuscript provides a detailed view of the bacterial community found on different Kefirs, I miss some studies about physiochemical and organoleptic features which allows to understand the role of the different bacteria found. Although commercial kefir was also analysed according to Material and Methods, results and discussion about this type of kefir was missed

Comments:

Line 16: Kefir shouldn’t be defined as a probiotic because although contains bacterial strains that belong to the same genus or species that strains which have been documented as probiotics, not a specific analysis have been undergone to determine the probiotic activity.

Line 59: Include the most updated reference for the probiotic concept: “Hill C. et al. 2014. The International Scientific Association for Probiotics and Prebiotics consensus statement on the scope and appropriate use of the term probiotics. Nat. Reviews Gastroenterol. Hepatol. 11(8):506-14.”

Line 59: “probiotic beverage”. Please don’t use the probiotic word (see my first comment)

Line 59-60: LAB species with documented probiotic properties “change by beneficial” (see my first comment)

Line 67: “In our recently published work…” Please add the reference

Line 79: Add the abbreviation for South Korea

Line 167-168: Include results obtained with commercial kefir in supplemental material.

The genus Swaminathania was identified only with sequencing in the regions V3 and V4 in all type of kefirs with a higher abundance in the product than in the grains. Why? Please include this result and discuss it in the Discussion section .

Line 189-191: No clear.. Please rewrite it. It seems that Ac pasteurianus and Glu. morbifer are the most abundant species in those kefirs but the most abundant species was Lb. kefiranof (line 186).

Line 235. Change dot by comma after L.acidophilus

Figure 2: Please add a cluster analysis of the samples  as it would be really useful to discuss the similarity among kefir of different origins

Figure 4: Instead of this Figure, a PCoA of beta diversity will be more informative.

Discussion: Avoid describe again numerical results in this section. Results should be only described in Results section (e.g.  lines 285-289; lines 300-303). Discussion also should include commercial kefir analysed in this work as a control and it would be interesting to discuss the differences with the artisanal ones.

Taxonomy: Names of species are not standardised in the text. Only use the complete name the first time that appear in the text. For genus abbreviation only use the first letter in capital (ie A pasteurianus instead of Ac pasteurianus, L. kefiri instead of Lb. kefiri

Author Response

Dear Reviewers 1, 2, and 3,

Thank you for your careful reviews of our article entitled “Bacterial populations in international artisanal kefirs.” We appreciate your helpful comments and suggestions. We believe that the second version of our manuscript is an improved version.

Reviewer 3:

The aim of this paper is to characterise the bacteria microbiota of 6 different artisanal Kefir. Although the manuscript provides a detailed view of the bacterial community found on different Kefirs, I miss some studies about physiochemical and organoleptic features which allows to understand the role of the different bacteria found. Although commercial kefir was also analysed according to Material and Methods, results and discussion about this type of kefir was missed

Comments:

Line 16: Kefir shouldn’t be defined as a probiotic because although contains bacterial strains that belong to the same genus or species that strains which have been documented as probiotics, not a specific analysis have been undergone to determine the probiotic activity.

Line 59: Include the most updated reference for the probiotic concept: “Hill C. et al. 2014. The International Scientific Association for Probiotics and Prebiotics consensus statement on the scope and appropriate use of the term probiotics. Nat. Reviews Gastroenterol. Hepatol. 11(8):506-14.”

Line 59: “probiotic beverage”. Please don’t use the probiotic word (see my first comment)

Line 59-60: LAB species with documented probiotic properties “change by beneficial” (see my first comment)

Line 67: “In our recently published work…” Please add the reference

Line 79: Add the abbreviation for South Korea

Line 167-168: Include results obtained with commercial kefir in supplemental material.

The genus Swaminathania was identified only with sequencing in the regions V3 and V4 in all type of kefirs with a higher abundance in the product than in the grains. Why? Please include this result and discuss it in the Discussion section .

Line 189-191: No clear.. Please rewrite it. It seems that Ac pasteurianus and Glu. morbifer are the most abundant species in those kefirs but the most abundant species was Lb. kefiranof (line 186).

Line 235. Change dot by comma after L.acidophilus

Figure 2: Please add a cluster analysis of the samples  as it would be really useful to discuss the similarity among kefir of different origins

Figure 4: Instead of this Figure, a PCoA of beta diversity will be more informative.

Discussion: Avoid describe again numerical results in this section. Results should be only described in Results section (e.g.  lines 285-289; lines 300-303). Discussion also should include commercial kefir analysed in this work as a control and it would be interesting to discuss the differences with the artisanal ones.

Taxonomy: Names of species are not standardised in the text. Only use the complete name the first time that appear in the text. For genus abbreviation only use the first letter in capital (ie A pasteurianus instead of Ac pasteurianus, L. kefiri instead of Lb. kefiri

Response to Reviewer 3:

The aim of this paper is to characterize the bacteria microbiota of 6 different artisanal Kefir. Although the manuscript provides a detailed view of the bacterial community found on different Kefirs, I miss some studies about physiochemical and organoleptic features which allows to understand the role of the different bacteria found. Although commercial kefir was also analysed according to Material and Methods, results and discussion about this type of kefir was missed

Thank you for letting us know that “you miss some studies about physicochemical and organoleptic features which allows to understand the role of different bacteria found”. Our previous publication (Sindi et al. 2020) “Antimicrobial Activity of Six International Artisanal Kefirs Against Bacillus cereus, Listeria monocytogenes, Salmonella enterica serovar Enteritidis, and Staphylococcus aureus, includes a table with products’ description. We are including the table here for your perusal.

Table 1. Artisanal kefir origin, source, grains’ description and products’ description

Kefir origin

Source

Grains’ description

Products’ description

Lithuania

Etsy Inc.

Cauliflower-like appearance, off-white to pale yellow, medium size (1-10 mm) and firm grains

Mild, smooth and not sour (sweet)

Ireland

Etsy Inc.

Soft, small size (>1 mm) grains

Mild, sweet and pleasant taste, smooth, sweet aroma, fresh and cheesy

The Caucuses region

Etsy Inc.

Cauliflower-like appearance, off-white to pale yellow, size 2-10 mm, firm, rubbery with smooth grains

Earthy, cheesy aroma and sour taste

South Korea

[19]

Soft, curling, size 2 -10 mm

Earthy, cheesy aroma and sour taste

Britain

Etsy Inc.

Cauliflower-like appearance, small to large size (2.5- 50 mm), rubbery, firm, smooth grains

Creamy, earthy, cheesy aroma, slightly sour

Fusion Tea

Amazon

Cauliflower-like appearance, off-white to pale yellow, mixed sizes (2-7 mm), firm, rubbery textured grains

Smooth, mild sour, creamy, pleasant, and fresh, sweet, yeasty aroma

We included four commercial kefir samples, Lifeway (plain), The Greek Gods (plain), Wallaby (organic plain), and Maple Hill (organic plain), in the 16S rRNA gene sequencing process as controls to test the accuracy of identification. We indicated in our original manuscript that “Commercial kefirs used in the study were determined to have the exact probiotic bacteria, both at the genus and species level as listed on their labels.” For the second iteration of our manuscript, we have submitted a supplemental figure that exhibits relative abundances of bacterial phyla and an aggregation of the 10 most abundant bacterial genera in the commercial kefirs. We hope that you find this figure informative. We would be glad to incorporate the figure into the manuscript if you would like us to do so.

Line 16: Kefir shouldn’t be defined as a probiotic because although contains bacterial strains that belong to the same genus or species that strains which have been documented as probiotics, not a specific analysis have been undergone to determine the probiotic activity.

In our manuscript, we tried to link the term “probiotic” to lactic acid bacterial species that have been described in various publications to have “probiotic” properties. However, we completely understand your comment. We replaced the word “probiotic beverage” with “fermented beverage” in the second iteration of our manuscript. However, whenever we cited a published paper in the manuscript, we adhered to the terms (probiotic, beneficial, health promoting, natural, etc.) that the authors used.

Line 59: Include the most updated reference for the probiotic concept: “Hill C. et al. 2014. The International Scientific Association for Probiotics and Prebiotics consensus statement on the scope and appropriate use of the term probiotics. Nat. Reviews Gastroenterol. Hepatol. 11(8):506-14.”

Thank you for your suggestion. The probiotic concept by Hill et al. (2014) is included in the manuscript as follows and the new reference is added to the list of references.  

From the text: Recently, The International Scientific Association for Probiotics and Prebiotics recommended that “the term probiotic be used only on products that deliver live microorganisms with a suitable viable count of well-defined strains with a reasonable expectation of delivering benefits for the wellbeing of the host” [13].

Line 59: “probiotic beverage”. Please don’t use the probiotic word (see my first comment)

As mentioned above, we tried to link the term “probiotic” to lactic acid bacterial species that have been described in various publications to have “probiotic” properties. However, we completely understand your comment. We replaced the word “probiotic beverage” with “fermented beverage” in the second iteration of our manuscript. However, whenever we cited a published paper in the manuscript, we adhered to the terms (probiotic, beneficial, health promoting, natural, etc.) that the authors used.

Line 59-60: LAB species with documented probiotic properties “change by beneficial” (see my first comment)

Thank you for your suggestion. We replaced the word “probiotic properties” with “beneficial properties” in the second iteration of our manuscript.

Line 67: “In our recently published work…” Please add the reference

Thank you for your suggestion. We added the following reference in the new list of references:

Sindi, A.; Badsha, M. B.; Nielsen, B.; Ünlü, G., Antimicrobial Activity of Six International Artisanal Kefirs Against Bacillus cereus, Listeria monocytogenes, Salmonella enterica serovar Enteritidis, and Staphylococcus aureus. Microorganisms 2020, 8 (6). DOI: 10.3390/microorganisms8060849.

Line 79: Add the abbreviation for South Korea

Thank you for your suggestion. We added K, the abbreviation for South Korean kefir.

Line 167-168: Include results obtained with commercial kefir in supplemental material.

We have submitted a supplemental figure that exhibits relative abundances of bacterial phyla and an aggregation of the 10 most abundant bacterial genera in the commercial kefirs.

The genus Swaminathania was identified only with sequencing in the regions V3 and V4 in all type of kefirs with a higher abundance in the product than in the grains. Why? Please include this result and discuss it in the Discussion section .

Thank you for your suggestion, we added the following statements to the results and discussion sections:

Lines 183-185: In addition, the V3-V4 region identified the genus Swaminathania in all kefir products and in all kefir grains (Figure 1-B), while it was not identified by the V1-V3 region (Figure 1-A). 

Lines 327-333: One interesting outcome of our study was the identification of the genus Swaminathania through the use of the V3-V4 region of the 16S rRNA gene but not through the use of the V1-V3 region. Since the 16S rRNA gene sequencing of four commercial kefirs, used as controls to test the accuracy of the identification, determined the exact probiotic bacteria listed on their labels, we are confident with the results of the 16S rRNA gene sequencing. The interesting outcome can be attributed to the fact that different variable regions in the 16S rRNA gene can be less or more suitable to identify bacteria at the genus or species level. Perhaps the V1-V3 region is less suitable for identification of the genus Swaminathania than the V3-V4 region.

Line 189-191: No clear.. Please rewrite it. It seems that Ac pasteurianus and Glu. morbifer are the most abundant species in those kefirs but the most abundant species was Lb. kefiranof (line 186).

Thank you for recognizing the unclear section. The following sentence has been added to the second iteration of the manuscript:  L. ultunensis was found to be more abundant in KP and KG when compared to all other artisanal kefirs and their grains. A. pasteurianus was more abundant (17.1%) in KP while G. morbifer was more abundant (5.8%) in LP when compared their abundance in all other artisanal kefirs.

Line 235. Change dot by comma after L.acidophilus

Thank you for your comment, however, those were two separate sentences. We changed the last sentence to make it clearer as follows:

Kefir K was a stand-alone kefir product with respect to the ranking of species with known bacteriocin production: L. helveticus, L. crispatus, Lent. kefiri, L. apis, and L. acidophilus. The most abundant LAB with known bacteriocin production capability was identified to be L. helveticus in KP with relative abundance of 21.8% and in KG with relative abundance of 17.8% (Figure 3).

Figure 2: Please add a cluster analysis of the samples as it would be really useful to discuss the similarity among kefir of different origins

We added a Hierarchical Clustering (HC) dendrogram (Figure 5), where the complete linkage method used for computing distance between clusters to determine similar clusters among international artisanal kefirs.

Figure 4: Instead of this Figure, a PCoA of beta diversity will be more informative.

Thank you for your comment. As per your request, we implemented the Principal Coordinates Analysis (PCoA) plots and submitted them as supplemental material. The PCoA does not appear to serve as a good interpreter of the significant correlation between a given kefir product and its grains.

The main purpose behind including Figure 4 was to obtain information about the strength and direction of a relationship between a given kefir product and its grains. The p-value was < 2.2e-16 from the correlation test, which indicates the correlation statistically significant. The fitted linear regression lines in each plot also confirm a very good fit of the models and p-value was < 2.2e-16. Also, both multiple R-squared and adjusted R-squared are higher than 0.95 in each plot. These results suggested statistically significant correlation between kefir product and its grains.

Discussion: Avoid describe again numerical results in this section. Results should be only described in Results section (e.g.  lines 285-289; lines 300-303). Discussion also should include commercial kefir analysed in this work as a control and it would be interesting to discuss the differences with the artisanal ones.

Thank you for your suggestion. We have removed any numerical result that is not tied to a discussion for the discussion section.

As mentioned above, we included four commercial kefir samples in the 16S rRNA gene sequencing process as controls to test the accuracy of identification. We indicated in our manuscript that “Commercial kefirs used in the study were determined to have the exact probiotic bacteria, both at the genus and species level as listed on their labels.” For the second iteration of our manuscript, we have submitted a supplemental figure that exhibits relative abundances of bacterial phyla and an aggregation of the 10 most abundant bacterial genera in the commercial kefirs. We hope you find this figure informative. We would be glad to incorporate the figure into the manuscript if you would like us to do so.

Taxonomy: Names of species are not standardised in the text. Only use the complete name the first time that appear in the text. For genus abbreviation only use the first letter in capital (ie A pasteurianus instead of Ac pasteurianus, L. kefiri instead of Lb. kefiri

Thank you for your suggestion. We tried to use as many single letter abbreviations as we could. However, we have many genera that share the same first letter. For example, the following genera all start with the letter L: Lacticaseibacillus, Lactococcus, Lactobacillus, Lentilactobacillus, Leuconostoc, and Listeria. We have differentiated all genera from each other in the manuscript not to create any confusion. We have used the following abbreviations in our manuscript. Please let us know if you would like to see another approach to differentiate the genera of interest.

Acetobacter (A.); Acidocella (Acid.); Bacillus (B.); Corynebacterium (C.); Gluconobacter (G.); Lacticaseibacillus (Lacti.); Lactococcus (Lac.); Lactobacillus (L.); Lentilactobacillus (Lenti.); Listeria (Lis.); Streptococcus (S.).

Round 2

Reviewer 1 Report

The paper can be accepted for publication

Reviewer 3 Report

Dear Authors,

Thank you very much for all the changes made to the manuscript as it has increased its quality. Although the manuscript is well-structured, it is just  a descriptive research about the microbiota composition of different types of kefirs and further studies should be perform to understand the role of the different bacteria beyond the antimicrobial properties. So from my point of view, this manuscript would fit better in a journal with less impact factor.

I would like also to suggest the revision of the nomenclature used along the text. Although different genera starts with the same letter, according to the international code of nomenclature of prokaryotes, only the first letter of the genus should be used in the abbreviation. According to this code if  species are listed belonging to two or more genera which have the same initial letter, the generic name should he used in full1

1  https://doi.org/10.1099/ijsem.0.000778